# Detection of carbapenem-resistant *Klebsiella pneumoniae* on the basis of matrix-assisted laser desorption ionization time-of-flight mass spectrometry by using supervised machine learning approach

Tsi-Shu Huang[1], Susan Shin-Jung Lee[1,2,3], Chia-Chien Lee[1], Fu-Chuen Chang[4]*

1 Division of Microbiology, Department of Pathology and Laboratory Medicine, Kaohsiung Veterans General Hospital, Kaohsiung, Taiwan, 2 Division of Infectious Diseases, Department of Internal Medicine, Kaohsiung Veterans General Hospital, Kaohsiung, Taiwan, 3 Faculty of Medicine, School of Medicine, National Yang-Ming University, Taipei, Taiwan, 4 Department of Applied Mathematics, National Sun Yat-sen University, Kaohsiung, Taiwan

* changfc@math.nsysu.edu.tw

**Data Availability Statement:** The data are all contained within the manuscript.

## Abstract

### Background

Carbapenem-resistant *Klebsiella pneumoniae* (CRKP) is emerging as a significant pathogen causing healthcare-associated infections. Matrix-assisted laser desorption/ionisation mass spectrometry time-of-flight mass spectrometry (MALDI-TOF MS) is used by clinical microbiology laboratories to address the need for rapid, cost-effective and accurate identification of microorganisms. We evaluated application of machine learning methods for differentiation of drug resistant bacteria from susceptible ones directly using the profile spectra of whole cells MALDI-TOF MS in 46 CRKP and 49 CSKP isolates.

### Methods

We developed a two-step strategy for data preprocessing consisting of peak matching and a feature selection step before supervised machine learning analysis. Subsequently, five machine learning algorithms were used for classification.

### Results

Random forest (RF) outperformed other four algorithms. Using RF algorithm, we correctly identified 93% of the CRKP and 100% of the CSKP isolates with an overall classification accuracy rate of 97% when 80 peaks were selected as input features.

### Conclusions

We conclude that CRKPs can be differentiated from CSKPs through RF analysis. We used direct colony method, and only one spectrum for an isolate for analysis, without modification

**Funding:** This work was financially supported by Kaohsiung Veterans General Hospital (VGHKS108-167 to TSH), https://eng.vghks.gov.tw/Default.aspx?r=1895294850.

**Competing interests:** The authors have declared that no competing interests exist.

of current protocol. This allows the technique to be easily incorporated into clinical practice in the future.

## Introduction

Carbapenem-resistant *Enterobacteriaceae* (CRE) is listed as one of the most urgent antibiotic resistance threats by the Centers for Disease Control and Prevention (CDC) and The World Health Organization (WHO) [1, 2]. Carbapenem resistance can be mediated by production of carbapenemases or by the combination of outer membrane porin expression disruption and production of various β-lactamases [3, 4]. Carbapenemases may confer resistance to virtually all available beta-lactam antibiotics. The most frequent carbapenemases are the *Klebsiella pneumoniae* carbapenemase (KPC) enzymes. KPC-producing bacteria are mostly in the form of *K. pneumoniae.* Since the first report of KPC-producing *K. pneumoniae* in the United States in 1996[5], KPC-producing organisms are now found in many other countries. [6–9]. In 2014, a global study reported 6.0%, 15.7%, 16.0%, 3.8% and 14.9% of meropenem nonsusceptible *K. pneumoniae* occurred in the United States, Europe, Latin American, APAC, and China [10]. The European Centre for Disease Prevention and Control (ECDC) report showed that between 2015 and 2018, there was a small but significant increasing trend of population-weighted mean percentages for carbapenem resistant *K. pneumoniae* (CRKP), from 6.8% to 7.5%, in the European Union (EU)/European Economic Area (EEA)[11]. The study of ECDC on the health burden of antimicrobial resistance estimated that the number of deaths attributed to infections with CRKP increased six-fold between 2007 and 2015. Even in countries with lower levels of CRKP, the impact of antimicrobial resistance on national burden is significant because of the high attributable mortality of these infections [12]. Therefore, rapid detection of carbapenem-resistance is important to guide the initial choice of antimicrobial treatment, as well as to prompt initiation of effective infection control measures. However, detection of carbapenem resistance using susceptibility testing methods are time consuming. Polymerase chain reaction (PCR) is a rapid, accurate and reliable method to detect resistant genes. However, it is limited to reference laboratories and can only detect known enzymes.

MALDI-TOF MS is now widely used to identify clinical microorganisms because of lowered cost and faster turn-around times [13, 14]. In recent years, four approaches have been reported for detection of certain drug resistant bacteria [15, 16] based on the postulation that any or combination of the polypeptides related to drug resistance could possibly alter the spectra obtained by MALDI-TOF MS and possibly enable differentiation from the drug-susceptible bacteria. One approach concerns the analysis of antibiotics and their degradation products [15, 17–19]. The hydrolysis degradation product of the β-lactam antibiotic shows a molecular mass different from that of the native molecule. The detection of β-lactamase activity was performed by analyzing the spectra containing peaks representing the β-lactam molecule, its salts (usually sodium salts), and/or its degradation products. For example, with MALDI-TOF MS, the presence of carbapenemases was confirmed by the detection of carbapenem hydrolysis (loss of molecular peaks: 476 5 m/z for ertapenem and its sodium salts, 498 5 m/z, 520 5 m/z; 383.0 m/z for meropenem and its sodium salts, 405.2 m/z and 427.4; 300.0 m/z for imipenem) and presence of their degradation products (450 m/z for ertapenem; 401.1 m/z for meropenem and its sodium salts, 423.3 m/z, 445.6 m/z, 467.839 m/z; and 254.0 m/z for imipenem). Moreover, the combination the β-lactam antibiotic with β-lactamase inhibitors has the potential of identification of the type of β-lactamase [20, 21]. The second approach directly detect β-

lactamase [22] A peak of approximately 29 kDa that represented β-lactamase was detected in the spectra. The third approach detects the specific peaks in the resistant group [23]. Some peaks were invariably present in, and unique to drug resistant organism or susceptible ones. Edwards-Jones V et al. reported some peaks unique to methicillin-resistant *Staphylococcus aureus* (MRSA)(e.g., 891, 1140, 1165, 1229 and 2127) or methicillin-susceptible *S.aureus* (MSSA)(e.g., 2548 and 2647). The m/z peaks 1774.1 and 1792.1 reported by Lu et al. were able to successfully discriminate between hospital-acquired MRSA and community-acquired MRSA isolates. The MALDI-TOF instruments acquire spectra in a linear positive-ion mode at a laser frequency of 60 Hz across a mass/charge (*m/z*) ratio of 2,000 to 20,000. Therefore, the first three approaches can detect molecules beyond the detection limit.

The fourth approach involves the analysis of ribosomal DNA methylations [24]. The methylation of rRNA confers resistance to antibiotics that inhibit protein synthesis. Methyltransferase activity was investigated by detection of digested rRNA to yield smaller products that were subsequently analyzed by MALDI-TOF MS. This approach requires specific sample preparations and purified ribosomes and enzymes, which also impedes its use in routine clinical laboratory. In addition, all these applications are limited by the need to know the specific target peak as a biomarker.

Some studies describe application of machine learning methods for differentiation of drug resistant from susceptible bacteria by directly using the profile spectra of whole cells MALDI-TOF MS. Griffin et al [25] successfully used a support vector machine (SVM) algorithm to accurately identify *vanB* positive, vancomycin-resistant *Enterococcus faecium* from vancomycin-susceptible isolates, directly using the profile spectra of whole cells. Although these preliminary results were promising, clonal relatedness of these strains was not clarified. It is therefore impossible to exclude the chance that the findings reflect only a favorable epidemiological condition in the geographical region of the authors. In addition, ethanol-formic acid extraction was used for sample preparation. It involves several centrifugation steps and is more labor-intensive than direct analysis of bacteria without additional protein extraction, which is usually done in routine work flow.

Mather et al. [26] successfully differentiated vancomycin-intermediate *S. aureus* (VISA) and heterogeneous VISA (hVISA) from vancomycin-susceptible *S. aureus* (VSSA) by using a SVM algorithm. A total of 20 spectra from biological duplicate experiments were acquired for each isolate. Only peaks present in 80% of the spectra for a given isolate ($\geqq$16/20) were selected and merged into one representative spectrum for further analysis. The peak selection criteria required multiple spectra for a given isolate and preprocessing for superspectrum made it difficult for use in clinical laboratory. Furthermore, the reproducibility of these findings has not been widely applied and published.

Supervised machine learning is a powerful tool for classification. However, MALDI-TOF mass spectrometry data consist of hundreds or thousands of mass to charge (m/z) ratio per specimen and an intensity level for each m/z ratio. The dimensionality is usually much larger than the sample size. This makes many standard pattern classification algorithms fail. Therefore, it is critical to reduce dimensions before discrimination using such data.

Here we describe an application of machine learning methods for differentiation of drug resistant bacteria from susceptible ones directly using the profile spectra of whole cells MALDI-TOF MS in 46 CRKP and 49 CSKP isolates. Data were preprocessed, consisting of peak matching and a feature selection step. Subsequently, five supervised machine learning algorithms were compared. We demonstrated that the RF algorithm outperformed other methods including logistic regression, naïve Bayes, nearest neighbors, and SVM. Using the RF algorithm, we correctly identified 93% of the CRKP and 100% of the CSKP isolates with an overall classification accuracy of 97% when 80 peaks were selected as input features.

## Materials and methods

### Bacterial strains

This study was approved by the Institutional Review Board (VGHKS18-CT8-07). All bacteria strains used in this study were from frozen stocks that was stored in 50% glycerol and/or Mueller-Hinton (MH) broth at −80˚C. Minimum inhibitory concentrations (MICs) of carbapenems were determined by Vitek 2 (bioMérieux, Marcy l'Etoile, France) test using the AST N320 card (bioMérieux, Durham, NC), which included imipenem (range, 1 to 16 μg/ml) and meropenem (range, 0.25 to 16 μg/ml), according to the manufacturers' recommendations. Study strains included in this study were *K. pneumoniae* isolates that were resistant to imipenem or meropenem according to the updated 2018 CLSI breakpoints [27], and had a "carbapenemase" phenotype detected by the advanced expert system (AES) of VITEK 2 system.

### MALDI-TOF MS analysis

**MALDI-TOF MS.** A portion of a colony was picked up from the blood agar plate and spotted onto a MALDI-TOF target plate. Each deposit on the target plate was overlaid with 2 μL of matrix solution (α-cyano-4-hydroxycinnamic acid) (VITEK® MS CHCA) and air dried.

Following sample preparation, samples were analyzed with the Vitek MS system (bioMérieux SA) in linear positive-ion mode, at a laser frequency of 50 Hz across the mass-to-charge ratio range of 2,000 to 20,000 Da. For each target slide, the *E. coli* reference strain ATCC 8739 [28] was used for instrument calibration according to the manufacturer's specifications. Quality of protein extraction was assessed by the data count (DC), defined by the interpretable number of peaks considered in the algorithm. These isolates with a score less than 90% were repeated once.

*Spectral analysis*. To recognize well-defined peaks, each spectrum was processed by baseline correction, denouncing, and peak detection. The data was transferred from the Vitek MS acquisition station to the Vitek MS RUO v4.12 after spectrum acquisition. The data were presented as a spectrum of intensity versus mass, in Daltons (Da).

**Peak matching.** Mass-Up software (http://sing.ei.uvigo.es/mass-up) was used to preprocess the data [29]. The list of peaks and intensities were imported to Mass-Up as CSV files. An inter-sample peak matching was performed with the Mass-Up software configured to use a tolerance of 300 ppm for forward algorithm. This step unifies the peak values among samples.

**Data analysis.** All of the following data analysis was performed using aligned spectra output from Mass-up software in mathematica software v. 12 (Wolfram Research, Champaign, IL). The performance of the machine learning prediction procedure was evaluated using a leave-one-out (L1O) cross-validation procedure. The overall flowchart and the representative steps of procedures of dimension reduction and cross validation steps were shown in Fig 1.

**Split data.** Each spectrum was 'left out' in turn as testing set, all the other spectra were training set. The testing set was not used in the training set. The dimension reduction process and building classifier was then executed using all the remaining data. The resulting machine learning structure was then tested using the left-out spectrum in the beginning. The process repeats, each time omitting a different spectrum, until all spectra have been omitted once.

**Dimension reduction.** The Student's t-test was used to compare the distribution of each m/z ratio (the null hypothesis H0 is that CRKP and CSKP have the same distribution) and generate a $p$ value for the $t$-statistic. Ranking the $p$ value from the smallest (most significant between two groups) to the largest. Select a list of top ranked $k$ peaks to machine learning algorithms to build classifier.

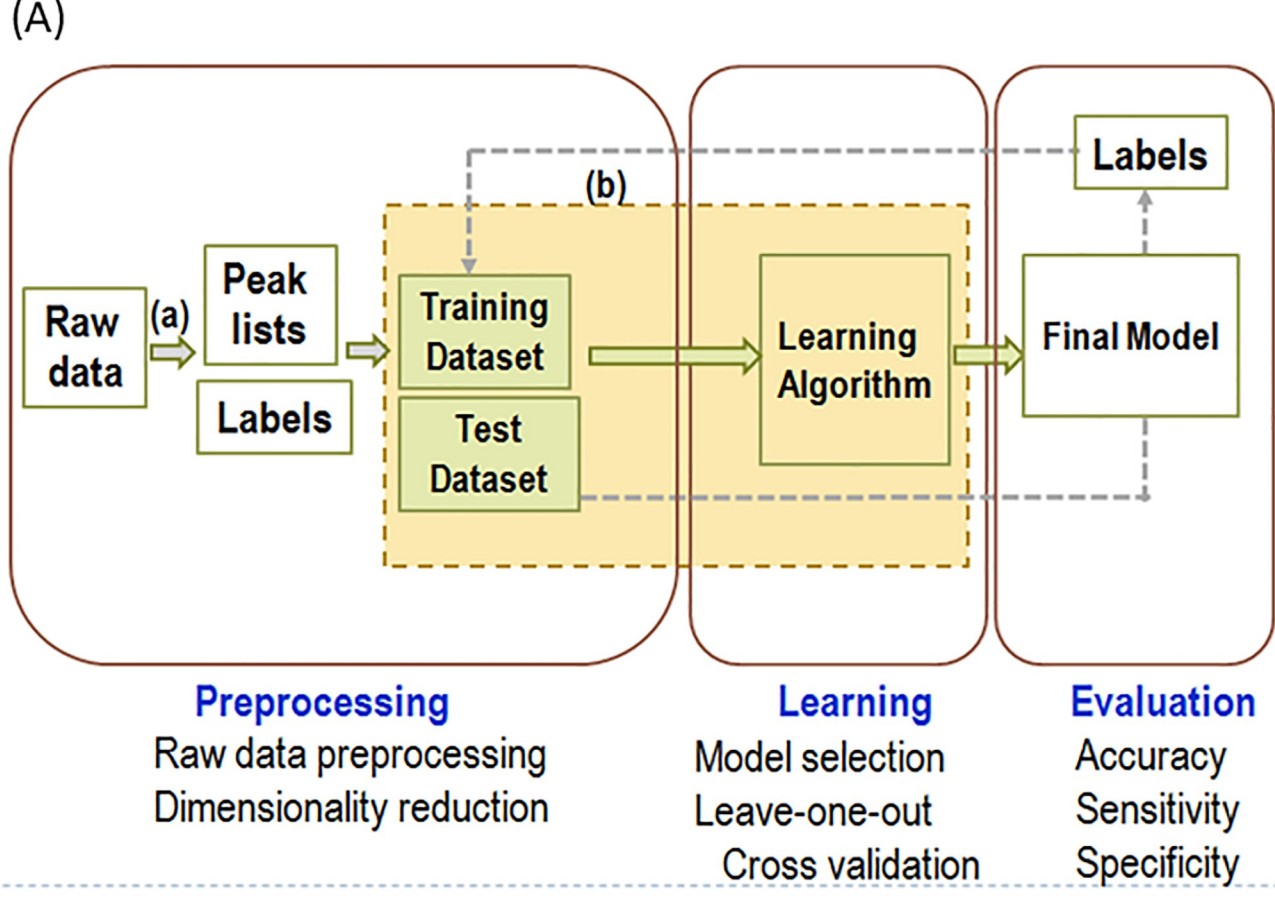

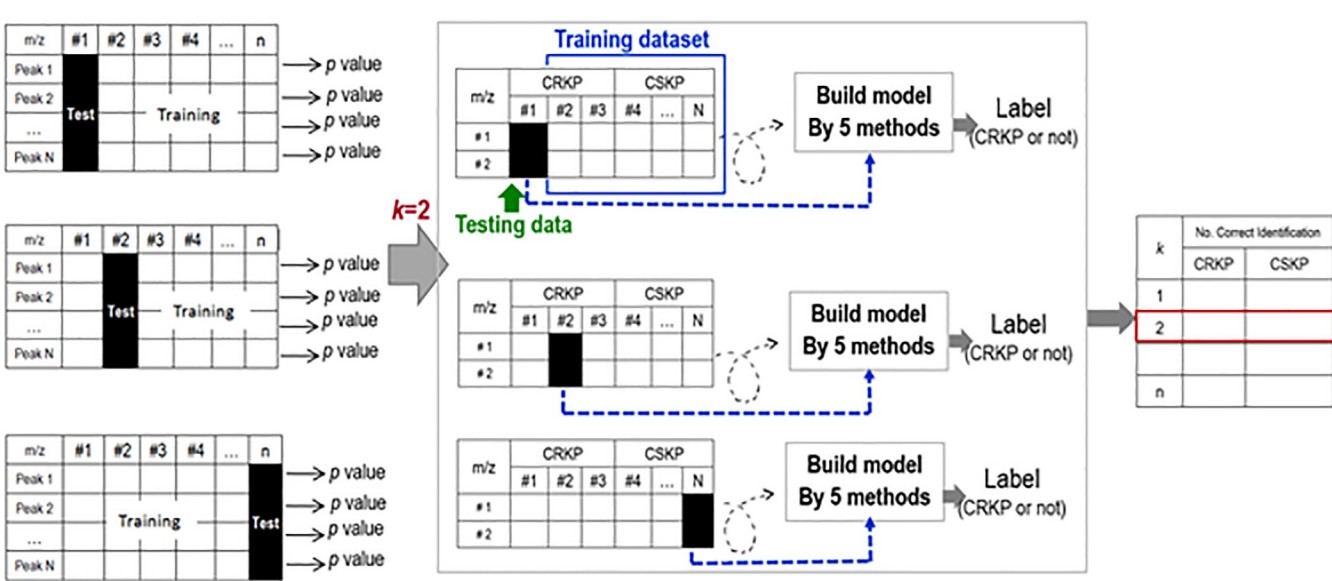

**Fig 1.** Overall flowchart (A) and the representative steps of procedures of dimension reduction and cross validation steps (B).

**Class prediction.** Five machine learning algorithms were performed for construction of classifiers: SVM, RF, nearest neighbors, naïve Bayes, and logistic regression. For each model, the default settings were left unaltered; for example, with the RF model, automatic detection was selected for the fraction of features to be randomly selected to train each tree, the maximum number of examples in each leaf, and the number of trees in the forest. In addition, the regularization parameter is set as 0.5.

In these analyses, different combinations of the most significant peaks with smaller *p* value by the Students's *t*-test were used to build the models. The testing spectrum that was "left out" in the beginning was then repeated with the same parameters in feature selection and feature extraction of the training set and input to classifiers. The classifier output was then compared to the known group of the spectrum, and a performance tally is recorded. The process repeats, each time omitting a different spectrum, until all spectra have been omitted once. Classifier performance was assessed by calculating the accuracy, sensitivity and specificity with which test observations were classified over all iterations of the L1O cross-validation process.

Sensitivity of the methods was assessed by calculating the percentage of CRKP that were correctly identified. Specificity was assessed by calculating the percentage of CSKP that were correctly identified. Accuracy was assessed by calculating the percentage of CRKP and CSKP that were correctly identified. The machine learning algorithm predictions were output as probabilities and exported to a comma-separated file.

**Random Amplified Polymorphic DNA (RAPD).** Bacteria were grown overnight on Columbia agar medium supplemented with 5% sheep blood. The bacterial DNA was prepared by QIAamp DNA Mini Kit (Qiagen Inc., Valencia, Calif., USA) according to the manufacturer's instructions. The primer used for PCR was AP4 [30].

The reaction mixture contained 10 mM Tris-HCl (pH 8.3), 1.8 mM MgCl2, 2 μM primer, 400 μM each deoxynucleoside triphosphate, 2.5 U of Super-Therm Gold DNA polymerase (JMR Holdings), and 2μL of DNA in a final volume of 25μL. Amplification was performed in a GeneAmp PCR 9700 thermal cycler (Perkin-Elmer). Initial denaturation was for 4 min at 94˚C. Amplification was performed with 50 cycles of 1 min at 94˚C, 2 min at 30˚C, and 2 min at 72˚C, with a single final extension step of 10 min at 72˚C. PCR products were separated by electrophoresis in a 1.5% agarose gel and stained with ethidium bromide.A CRKP strain, S, is included in all runs to ensure comparability between different runs.

## Results

### Bacteria strains

A total of 46 CRKP strains isolated consecutively from January 2016 to October 2017 were selected from frozen stock, one isolate for each patient. Bacteria isolates included in this study were isolated from clinical specimens including 7 (7.4%) pus/wound, 5 (5.3%) sputum, 20 (21.1%) urine, 54 (56.8%) blood, and 5 (5.3%) body fluids like ascitic fluid, bile, pleural fluid and 4 (4.2%) other specimens like catheter tips. There was no nosocomial outbreak of CRKP during the study period. The distribution of cases by month ranged from 0 to 5, with a mean of 2.1 cases per month (standard deviation, 1.4). Fifty CSKP isolates were randomly selected over the same period, 49 of them successfully sub-cultured and included in this study.

Among the 46 CRKP, 29 (63.0%) were detected as "carbapenemase (+ or–ESBL)", 17 (37.0%) were carbapenemase (+ or–ESBL) plus "impermeability CARBA (+ESBL or +HL AmpC)". Phenotypes of the 49 non-CRKP isolates were one or combinations of wild (penicillinase), acquired penicillinase + impermeability (cephamycins), acquired cephalosporinase (except ACC-1), acquired penicillinase, ESBL (CTX-M Like), ESBL+ impermeability (cephamycins), extended spectrum β-lactamase, or inhibitor resistant penicillinase (IRT or OXA).

**Table 1. Top 10 peaks with least sum of 95 ranks produced each time one different spectrum omitted and their percentage of presence in each group.**

| m/z | Present in no. (%) of: | | No. of times the peak ranked as: | | | | | | | | | | |
|---|---|---|---|---|---|---|---|---|---|---|---|---|---|
| | CRKP | CSKP | 1 | 2 | 3 | 4 | 5 | 6 | 7 | 8 | 9 | 10 | >10 |
| 9478.866 | 38 (82.6%) | 1 (2.2%) | 95 | 0 | 0 | 0 | 0 | 0 | 0 | 0 | 0 | 0 | 0 |
| 9541.405 | 35 (76.1%) | 1 (2.2%) | 0 | 91 | 2 | 1 | 0 | 0 | 0 | 0 | 0 | 0 | 1 |
| 6288.794 | 35 (76.1%) | 12 (26.1%) | 0 | 2 | 67 | 26 | 0 | 0 | 0 | 0 | 0 | 0 | 0 |
| 7705.009 | 37 (80.4%) | 1 (2.2%) | 0 | 1 | 25 | 66 | 1 | 0 | 0 | 0 | 0 | 0 | 2 |
| 7158.634 | 41 (89.1%) | 12 (26.1%) | 0 | 1 | 1 | 1 | 91 | 0 | 0 | 0 | 0 | 0 | 1 |
| 10287.76 | 35 (76.1%) | 2 (4.3%) | 0 | 0 | 0 | 1 | 1 | 92 | 0 | 0 | 0 | 0 | 1 |
| 4768.279 | 25 (54.3%) | 10 (21.7%) | 0 | 0 | 0 | 0 | 2 | 1 | 89 | 3 | 0 | 0 | 0 |
| 2636.88 | 24 (52.2%) | 0 | 0 | 0 | 0 | 0 | 0 | 2 | 2 | 70 | 15 | 6 | 0 |
| 4362.217 | 21 (45.7%) | 10 (21.7%) | 0 | 0 | 0 | 0 | 0 | 0 | 2 | 8 | 58 | 18 | 9 |
| 5379.418 | 25 (54.3%) | 10 (21.7%) | 0 | 0 | 0 | 0 | 0 | 0 | 0 | 5 | 7 | 54 | 29 |

## Analysis of CRKP and CSKP proteomic profiles

Protein extraction was assessed with data count (DC) rated by Saramis v4.12. None of these peaks was distinctive to the CRKP or CSKP group. There was one peak (9478.87 Da) present in 82.6% of the CRKP isolates and only 2.2% of the CSKP isolates, one peak (7705.009 Da) was present in 80.4% of the CRKP isolates and only 2.2% of the CSKP isolates, and the other peak (9541.41 Da) was present in 76.1% of the CRKP isolates and only 2.2% of the CSKP isolates (Table 1).

The Student's t-test was used as the feature selection method on the basis of leave-one-out cross-validation. Student's t-test was used to compare the distribution of each m/z ratio and generate a *p* value for the *t*-statistic for each peak. The low *p* values obtained for each peak indicate that the observed difference in intensity of the individual peaks is highly statistically significant and not coincidence-based (the lower the p value, the better a respective peak signal is suitable for separating the two classes).

After removing one sample at a time, peaks were ranked according to the *p*-value calculated by the Students t-test on the remaining samples. The top *k* peaks were selected to construct the classifier. The process repeats, each time omitting a different spectrum, until all spectra have been omitted once. Therefore, 95 ranks were produced for each peak, equating to the total number of samples. As shown in Table 1, the peak of 7705.009 Da ranked first, no matter which sample was left out. This peak was highly significantly different between CRKP and CSKP isolates.

We evaluated the classification performance of the five machine learning algorithms for detection of CRKP with leave-one-out cross-validation, where the classifier was constructed after serial removal of a sample from the dataset with subsequent prediction for the removed sample with different feature subset when *k* = 1 to 100. As criteria, precision, sensitivity, and specificity were used and compared to estimate model validation. As shown in Fig 2, panel A, except using naïve Bayes algorithm, the maximum classification accuracy ranged from 89% to 97% when up to100 peaks were selected.

Maximal values for all three statistical parameters were greater than 90% except results obtained using naïve Bayes algorithm. RF outperformed other four algorithms with all three statistical parameters greater than 90% and their minimal values not less than 75% when 1–100 peaks were selected as input variables. The ranges of accuracy, sensitivity, and specificity were 81%-97%, 76%-96%, and 82%-100%, respectively, when 1–100 peaks selected. Classification results were influenced by not only the machine learning algorithm, but also by the

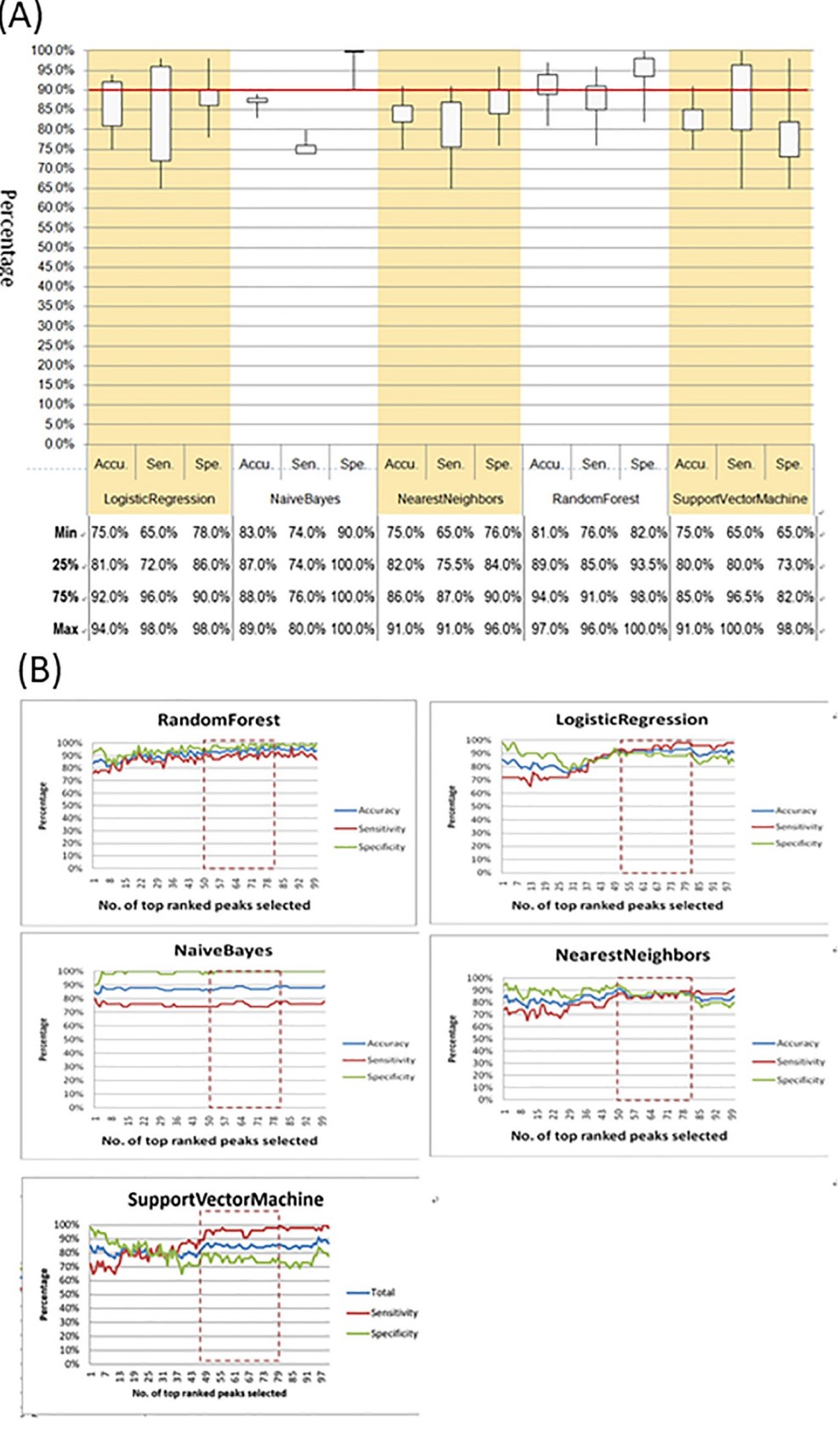

| | Accu. | Sen. | Spe. | Accu. | Sen. | Spe. | Accu. | Sen. | Spe. | Accu. | Sen. | Spe. | Accu. | Sen. | Spe. |
|---|---|---|---|---|---|---|---|---|---|---|---|---|---|---|---|
| | LogisticRegression | | | NaiveBayes | | | NearestNeighbors | | | RandomForest | | | SupportVectorMachine | | |
| Min | 75.0% | 65.0% | 78.0% | 83.0% | 74.0% | 90.0% | 75.0% | 65.0% | 76.0% | 81.0% | 76.0% | 82.0% | 75.0% | 65.0% | 65.0% |
| 25% | 81.0% | 72.0% | 86.0% | 87.0% | 74.0% | 100.0% | 82.0% | 75.5% | 84.0% | 89.0% | 85.0% | 93.5% | 80.0% | 80.0% | 73.0% |
| 75% | 92.0% | 96.0% | 90.0% | 88.0% | 76.0% | 100.0% | 86.0% | 87.0% | 90.0% | 94.0% | 91.0% | 98.0% | 85.0% | 96.5% | 82.0% |
| Max | 94.0% | 98.0% | 98.0% | 89.0% | 80.0% | 100.0% | 91.0% | 91.0% | 96.0% | 97.0% | 96.0% | 100.0% | 91.0% | 100.0% | 98.0% |

**Fig 2. Performance of five machine learning algorithms with leave-one-out cross validation using 46 CRKP and 49 CSKP mass spectra, in terms of accuracy, sensitivity and specificity for differentiate CRKP from CSKP.**
Accuracy (Accu), sensitivity (Sen) and specificity (Spe) are used to evaluate prediction systems. Panel A, Boxplots (25th to 75th percentiles, Min to max) of accuracy, sensitivity, and specificity for differentiation of CRKP from CSKP when 1–100 peaks selected. In Panel B, Values of accuracy, sensitivity and specificity with number of ranked peaks ($k$ = 1–100) with increasing $p$-value (X axis) selected for machine learning algorithms. $k$: number of ranked peaks selected with increasing $p$-value for classification by using all five machine learning algorithms.

number of peaks selected. We found that sensitivity was improved and variability decreased when 50 to 80 peaks were selected as features for building classifier, irrespective of which algorithm was used.

In Fig 2, panel B, while increasing the number of peaks selected as input variables, the classifier robustness increased. RF exhibited satisfactory performance. Results of classification performance were shown in Table 2. The maximal overall performance achieved when 80 peaks was selected by using RF. When 80 peaks selected as input features, classification sensitivity, specificity and accuracy achieved 93%, 100% and 97%, respectively.

There was one CRKP repeatedly misclassified by RF algorithm despite varying the number of peaks selected as input features. However, when Logistic Regression, Nearest Neighbors or SVM algorithm was used, it can be classified correctly all the time.

## Strain relatedness

Clonal relatedness of the study strains were clarified using RAPD fingerprinting method (Fig 3). The distribution of CRKP strains by month ranged from 0 to 5, with a mean of 2.1 cases per month. We found that RAPD profiles of CRKP strains showed different patterns. Thus, our machine learning model's classification of the CRKP and CSKP does not seem to be due to variations in strain type.

**Table 2. Performance of five machine learning algorithms with L1O cross validation using 46 CRKP and 49 CSKP mass spectra, in terms of accuracy, sensitivity and specificity for differentiate CRKP from CSKP.**

| Algorithm | Metric | No. of ranked peaks selected with increasing *p*-value | | | | | |
|---|---|---|---|---|---|---|---|
| | | **50** | **60** | **70** | **80** | **90** | **100** |
| Random Forest | Accuracy | 94% | 94% | 95% | 97% | 95% | 94% |
| | Sensitivity | 91% | 91% | 91% | 93% | 89% | 87% |
| | Specificity | 96% | 96% | 98% | 100% | 100% | 100% |
| Logistic Regression | Accuracy | 93% | 92% | 92% | 93% | 91% | 91% |
| | Sensitivity | 93% | 93% | 96% | 98% | 96% | 98% |
| | Specificity | 92% | 90% | 88% | 88% | 86% | 84% |
| Naïve Bayes | Accuracy | 86% | 88% | 87% | 89% | 88% | 89% |
| | Sensitivity | 74% | 76% | 74% | 78% | 76% | 78% |
| | Specificity | 98% | 100% | 100% | 100% | 100% | 100% |
| Nearest Neighbors | Accuracy | 91% | 84% | 87% | 87% | 83% | 85% |
| | Sensitivity | 87% | 83% | 87% | 89% | 87% | 91% |
| | Specificity | 94% | 86% | 88% | 86% | 80% | 80% |
| Support Vector Machine | Accuracy | 87% | 84% | 84% | 86% | 85% | 87% |
| | Sensitivity | 96% | 96% | 96% | 100% | 98% | 98% |
| | Specificity | 80% | 73% | 73% | 73% | 73% | 78% |

Machine learning analysis to identify carbapenem-resistant *Klebsiella pneumoniae*

**Fig 3. Random amplified polymorph in DNA fingerprinting (RAPD) types of 46 carbapenem-resistant *K. pneumoniae* generated by arbitrarily primed PCR.** Lanes S, standard strain included in every experiment as a control. Results for the study strains were shown in the order of the date of isolation. Lane M shows the 1-kb DNA ladder.

## Discussion

In this study, we demonstrated that MALDI-TOF MS can reliably differentiate CRKP from CSKP. We were able to achieve 97% classification accuracy of CRKP and CSKP isolates by MALDI-TOF MS spectral files using RF algorithm when 80 top ranked peaks were selected to input into machine learning algorithms to build the classifier.

MALDI-TOF MS has focused on the existence or lack of particular spectral peaks to recognize antibiotic-resistant bacteria in the previous reports [15–23]. Unlike most studies in which MALDI-TOF MS was used to predict antibiotic resistance, we did not focus on target peaks from enzymes or metabolites related to drug resistance. In this study, there were only three peaks in CRKP and CSKP isolates, which were not exclusive to either group. Whether these target peaks are potential biomarkers require further study which analyze and process large series of bacterial strains. The proposed method herein can be adapted to other classification problems by using MALDI-TOF data. This technology has the potential for broad application to detection of other drug resistance, genotyping, or any classification problems.

Many bioinformatics modeling tasks, such as sequence analysis over microarray analysis or spectral analysis, have high dimensional characteristics. Computation of hundreds of peaks from such tasks is a challenge for applying machine learning methods. To benefit the predictability of the machine learning process as well as the computation speed, it is essential to reduce dimensionality through feature selection. We use Student $t$-test to compare the distribution of each peak in CRKP and CSKP groups and generate a $p$ value for the $t$-statistics on the basis of L1O cross validation. Significant peaks with top ranking according to the $p$-value were selected as input features to construct classifier. In addition to reducing the dimensionality, it is also beneficial for reducing the effect of interference peaks obtained from culture or MALDI-TOF process. Also, the basal level of signal that is fundamentally not related with drug resistance will be excluded.

Due to limited availability of CRKP isolates, our study has the limitation of a relatively small sample size. We used a L1O cross validation procedure to assess the predictive ability of different learning algorithms. The tested data is not part of the training set used in the building model. The training set in each L1O iteration is different. The set of peaks selected in feature selection may vary from one iteration to another. We tested all available data with this method and used the largest possible training set in each trial. Furthermore, due to the re-sampling procedure in RF, the use of external samples may not be required to validate the prediction model [31].

If in each of the five algorithms the same isolates were consistently incorrectly classified, then this would indicate that they were most likely misidentified. This was not the case in our

study, the isolates that were incorrectly classified in one algorithm were correctly identified using at least one other algorithms. This would implicate that this is a reflection of the varying power between the different algorithms.

The combination of two methods, bagging (a contraction of bootstrap-aggregating) and the Classification And Regression Trees (CART)-split criterion [32,33], makes the RF a very effective way to build models of classification that are highly predictive. Bagging generates bootstrap samples from the original data set, constructs a predictor from each sample, and decides by voting for classifications [34, 35]. The bootstrap-aggregating procedure yields better model performance as it reduces the model's variance without increasing the bias. This means that while a single tree's predictions in its training set are sensitive to noise, the vote of many trees is not, as long as the trees are not correlated. Training many trees on a single training set would give strongly correlated trees; bootstrap sampling is a way to de-correlate trees by showing them different training sets. In RF, thousands of tree-like models are grown on bootstrapped data samples. Tree-like models split the data into groups repeatedly by the predictor variable and value leading to the most homogeneous post-split groups. RF further de-correlate the tree-type models by allowing each tree to choose on each split only from a small sub-selection of predictors. This way, each tree will fit some of the true patterns in the data, and some noise, that is unique to its bootstrap sample. The noise should balance out when the projections of all trees are averaged, and only the true signal remains. Therefore, RF can used when the number of variables is much larger than the number of observations, and returns measures of variable importance and is suitable for spectral analyses.

We also ensured that our findings were not simply due to clonal relatedness in the study strains. Random amplified polymorphic DNA (RAPD) method is used for the molecular epidemiology of ESBL-producing *K. pneumoniae* strains in a previous study (31). RAPD analysis is significantly simpler to perform and produces results more rapidly. Interassay reproducibility has been criticized with RAPD typing because it can be affected by minor variation in factors influencing regular PCR. Therefore, a control strain was included in each run to ensure inter-assay reproducibility.

We used the direct colony method which involves spotting a small amount of bacteria directly onto the target plate without extraction procedures. The same spot can be analyzed not only for identification purpose, but also add-on information of drug resistance. Furthermore, only one spectrum for an isolate was use to analysis. These benefits make it possible to incorporate this method into clinical practice without altering the current protocol.

Early detection of multiple drug-resistant microorganisms has the ability to enhance clinical decision making and patient outcomes. Although additional studies are required before this test is integrated into the normal workflow in clinical environments, we demonstrated the ability of machine learning methods to identify CRKP.

## Supporting information

**S1 Fig. Original gel images contained in the manuscript's Fig 3.**
(PDF)

## Author Contributions

**Data curation:** Tsi-Shu Huang, Chia-Chien Lee.

**Formal analysis:** Fu-Chuen Chang.

**Funding acquisition:** Tsi-Shu Huang.

**Investigation:** Tsi-Shu Huang, Fu-Chuen Chang.

**Methodology:** Chia-Chien Lee, Fu-Chuen Chang.

**Project administration:** Tsi-Shu Huang.

**Resources:** Fu-Chuen Chang.

**Validation:** Tsi-Shu Huang, Fu-Chuen Chang.

**Writing – original draft:** Tsi-Shu Huang.

**Writing – review & editing:** Susan Shin-Jung Lee, Chia-Chien Lee, Fu-Chuen Chang.

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
