## [Decision Letter · Decision Letter 0]

20 Nov 2019

PONE-D-19-26377

Detection of carbapenem-resistant Klebsiella pneumonia on the basis of matrix-assisted laser desorption ionization time-of-flight mass spectrometry by using supervised machine learning approach

PLOS ONE

Dear Professor Chang,

Thank you for submitting your manuscript to PLOS ONE. After careful consideration, we feel that it has merit but does not fully meet PLOS ONE’s publication criteria as it currently stands. Therefore, we invite you to submit a revised version of the manuscript that addresses the points raised during the review process.

We would appreciate receiving your revised manuscript by Jan 04 2020 11:59PM. To enhance the reproducibility of your results, we recommend that if applicable you deposit your laboratory protocols in protocols.io, where a protocol can be assigned its own identifier (DOI) such that it can be cited independently in the future. For instructions see: http://journals.plos.org/plosone/s/submission-guidelines#loc-laboratory-protocols

We look forward to receiving your revised manuscript.

Kind regards,

Joseph Banoub, Ph,D., D. Sc.

Academic Editor

PLOS ONE

Journal Requirements:

Additional Editor Comments:

Your manuscript will be reconsider after you complete the major revision suggested by referee 1.

Reviewers' comments:

Reviewer's Responses to Questions

**Comments to the Author**

1. Is the manuscript technically sound, and do the data support the conclusions?

Reviewer #1: Partly

Reviewer #2: Partly

2. Has the statistical analysis been performed appropriately and rigorously? 

Reviewer #1: No

Reviewer #2: I Don't Know

3. Have the authors made all data underlying the findings in their manuscript fully available?

Reviewer #1: Yes

Reviewer #2: Yes

4. Is the manuscript presented in an intelligible fashion and written in standard English?

Reviewer #1: No

Reviewer #2: Yes

5. Review Comments to the Author

Reviewer #1: Review of: Detection of carbapenem-resistant Klebsiella pneumonia on the basis of matrix assisted laser desorption ionization time-of-flight mass spectrometry by using supervised machine learning approach

The authors describe a machine learning algorithm that can differentiate between carbapenem resistant K. pneumonia and carbapenem sensitive K. pneumonia. Whole cells were analyzed with MALDI-ToF. While the manuscript could be of potential interest to readers, it needs major revisions to put it in its final form as outlined below.

Major comments:

1- The conclusion in the abstract section is vague. What do the authors mean by “analyzed spectra with the same MALDI-TOF MS settings used for bacterial identification. Furthermore, only one spectrum for an isolate was use to analysis.”

2- The introduction section is superficial and requires thorough revision in terms of the clarity of used language. Sentences are hard to read, although they describe mostly general information. The depth of the content, in my opinion, is not suitable for the audience. Some sections in the introduction also requires the addition of citations to support the authors’ claims. Furthermore, the compilation of the literature review is very confusing. I am not sure what the authors were trying to explain with the four approaches that identify drug resistant bacteria. There was no adequate description of what these approaches do in terms of drug resistant bacteria and how is the approach comparable to the presented work. Also no clear description of the used techniques in these approaches and the associated problems.

3- The authors did not attempt any peak identification at this stage, neither did they correct for isotope, adducts and salts during peak annotation. As such, I don’t see the validity in assessing the significance of the difference in the number of peaks between CSKP and CRKP. As such, figure 2 and discussions in the manuscript on that topic should be removed.

4- The authors selected 80 top ranked unidentified peaks to differentiate between CRKP and CSKP? Can the authors comment on the uniqueness of these peaks to solely carbapenem resistance and whether these peaks are also found in other penicillins that klebsiella is also resistant to such as amoxicillin and ampicillin?

5- Why didn’t the authors attempt for the identification of the peaks, especially with the m/z 9478.87 and 7705.009 as being mostly present in CRKP. Can the authors comment on the availability of data bases.

6- How was the feature selection using t-test conducted, i.e. what was the software used for it? Why did the authors use a univariate statistical test for feature selection? Why multivariate statistical testing such as PCA was not used?

Minor comments:

1- Line 26: “healthcare-associated pathogen” do the authors mean it infect healthcare professionals?

2- Authors should state the full name of the acronym at its first appearance throughout the document, e.g. ToF, CDC, WHO, PCR, MIC, CHCA and RF

3- Line 34: description of the utility of MALDI-ToF should be a part of the introduction and not the methods section of the abstract.

4- Line 40: authors already defined CSKP

5- Line 54: could the authors explain more on the urgency of carbapenem resistance as a threat to healthcare?

6- Line 56: do the authors mean that usually involve a combination of mechanisms at a time, the authors stated once mechanism? Also what is the role of B-Lactamase inhibitors as a combination therapy in addressing this resistance?

7- Line 58: what is KPC?

8- Line 58? What do the authors mean by “KPC-producing bacteria have increasingly been isolated worldwide” this is a very general term and not suitable for a scientific audience.

9- Line 68: the sentence needs to be thoroughly revised to indicate that polypepetides related to drug resistance could alter the spectra usually obtained from susceptible strains.

10- Line 74: what do the authors mean by: “The first three approaches usually detect small molecules (<74 1,000 Da) beyond the detection limit of the MALDI-TOF instruments now commonly employed in clinical microbiology, which spectra were acquired in a linear positive-ion mode at a laser frequency of 60 Hz across a ass/charge (m/z) ratio of 2,000 to 20,000.”

11- Line 89: can the authors comment on the advantages of analyzing whole cells over solvent extraction?

12- Line 109 and line 157: please use SVM abbreviation instead.

13- Line 122: can the authors comment on the possibility of carabapenemase phenotype susceptibility to meropenem or imipenem?

14- Line 125: authors need to be more specific in describing their specimen size. How much is a portion?

15- Line 127: did the authors do any optimization in regards to the MALDI matrix? Was there any matrices tried?

16- Line 131: can the authors reference the manufacturer specification that describes the use of E. Coli as a reference strain?

17- Line 133: I am not clear on the selection criteria for ensuring the quality of the protein extraction. Was 90% a score that was set in-house for assessing the quality of protein extraction or it is a common practice? When isolates were re-analyzed, were the scores from the repeats only used in the analysis or the scores before and after repeating were averaged? In general, was it an n=1 that was used throughout the study?

18- Line 134: did the authors correct for isotopes and salts in their spectral analysis?

19- Line 135: what was the software used for spectral analysis?

20- Line 151: did the authors apply any corrections for the multiple student t-testing?

21- Line 176: was the temperature set at 208 C?

22- Line 188: where were the isolates collected from the patient, are they respiratory fluids? Urine?

23- is there an ethics approval for collecting patient samples?

24- Line 202? What is the unit of dimensionality?

25- Line 245: the authors stated that random forest algorithm outperformed the other techniques, however there was one CRKP that was constantly misclassified using the random forest algorithm while was correctly classified using the other techniques? Can the authors comment on this finding? Did they run any blinded samples to confirm the superior performance of one algorithm over the other?

26- Line 264: the authors mentioned that they did not focus on targeted peaks from enzymes or metabolites related to drug resistance. However, they detected 3 peaks that were differentially present in CRKP and CSKP. Can the authors comment on the significance of their findings in regards to biomarker discovery?

27- Line 306: the authors explicitly detailed the advantages of random forest testing, however, they did not justify the inclusion of other algorithms in this study.

28- Figure 1 and figure 4 were not cited in text

29- Can the authors make a clear discrimination between the specification of the training set and the test set?

30- Figure 3. Can the authors specify what is on the x and y axis in panel A and B?

31- Reference formatting should be corrected for errors

Reviewer #2: In this Manuscript, the authors used MALDI-TOF-MS to differentiate between Carbapenem-resistant and Carbapenem-susceptible Klebsiella pneumonia pathogens (CRKB and CSKB). The authors used statistical techniques to analyze the MALDI-MS data of 46 CRKB and 49 CSKB and to build classifiers for differentiation purposes between these two types of pathogens.

The classifiers MS peaks are shown in table 1. From this list, they found that “ there were none of these peaks distinctive to CRKP or CSKP, although one peak (9478.87 206 Da) was present in 82.6% of the CRKP isolates and only 2.2% of the CSKP isolates, one peak (7705.009 Da) was present in 80.4% of the CRKP 207 isolates and only 2.2% of the CSKP isolates, 208 and the other peak (9541.41 Da) was present in 76.1% of the CRKP isolates and only 2.2% of 209 the CSKP isolates.”

Overall, the work sounds technically good, but it needs statistics experts to assess these statistical techniques used to build these classifiers peaks.

6. PLOS authors have the option to publish the peer review history of their article (what does this mean?). If published, this will include your full peer review and any attached files.

Reviewer #1: No

Reviewer #2: No

---

## [Author Response · Author response to Decision Letter 0]

14 Jan 2020

To the Editor

Plos ONE Editorial Office

Re: PONE-D-19-26377

Thank you for your constructive review of this manuscript. My co-authors appreciate the opportunity to address the reviewer's comments and to revise the manuscript accordingly.

Reviewer #1

Major comments:

1. The conclusion in the abstract section was rewritten. 

2. The introduction was revised to state the problem more clearly and explain why it is important and why we chose the study method. We also made a brief review of the previous studies, including what these approaches do in terms of drug resistant bacteria, the techniques they used, and how these approaches compares to the presented work.

3. Figure 2 and discussions in the manuscript on that topic was removed, as suggested by the reviewer.

4. Klebsiella spp. are intrinsically resistant to penicillins, i.e. all of the CRKP as well as CSKP are penicillin resistant. Therefore, the peaks that differentiate between CRKP and CSKP are not related to penicillins, since both CRKP and CSKP are resistant to penicillin.

5. It is not able to find potential peptides/proteins of interest from just m/z ratio alone. We demonstrated that CRKPs can be differentiated from CSKPs through supervised machine learning method. It is possible to consider the specific attributes (such as m/z ratio) as biomarkers for the defined classes. Whether these target peaks are potential biomarkers require further study which analyze and process large series of bacterial strains (Line 296-298).

6. Feature selection using t-test was performed using Mathematica software v. 12 (Wolfram Research, Champaign, IL) (Line 173-174). Both t-test and PCA essentially reduce the dimensionality of the data, however, the overall performance by using PCA was inferior to t-test (accuracy was 85-92%, with 74%-83% of sensitivity and 92%-100% of specificity). 

Minor comments:

1. The sentence was corrected as “a significant pathogen causing healthcare-associated infections”. (Line 26).

2. The full name of the acronyms at its first appearance were added: ToF (Line 27) CDC WHO (Line 49-50) PCR (Line 70) MIC (Line 146-147) CHCA (Line 157) RF (Line 109).

3. The abstract was rewritten.

4. Full name of CSKP Deleted.

5. The urgency of carbapenem resistance as a threat to healthcare was described in Introduction (Line 52-66). 

6. This sentence was corrected as “Carbapenem resistance can be mediated by production of carbapenemases or by the combination of outer membrane porin expression disruption and production of various β-lactamases” (Line 50-52). The role of β-lactamase inhibitors combined with β-lactam antibiotic to address the resistance was added. (Line 88-89).

7. The full name of KPC was added. (Line 55)

8. The epidemiology of carbapenem-producing K. pneumoniae was described in Introduction (Line 55-63).

9. We rewrote the paragraph to indicate why spectral differences between resistant and sensitive isolates are expected (Line 78-97).

10. The first three approaches usually detect molecules beyond the detection limit of the MALDI-TOF instruments now commonly employed in clinical microbiology were explained more clearly (Line 78-100). 

11. Solvent extraction involves several washing and centrifugation steps and is more labor-intensive than direct analysis of bacteria without additional protein extraction, which is usually done in routine work flow (Line 116-118).

12. SVM abbreviation was used. 

13. The carbapenemase phenotype in the Vitek2 AES in our hospital predicts imipenem and ertapenem susceptibility, but not meropenem. Meropenem was not included in our panel (Line 145-149). 

14. Sample size was specified as approximately three millimeters in diameter, according to the User Manual VITEK MS Clinical Use-Workflow) (Line 154).

15. All the procedures and reagents were conducted according to the manufacture’s recommendations for identification of bacteria in a routine laboratory without any modification. 

16. The Reference of E. coli as reference strain was cited (Line 161).

17. The score 90% is a common practice. When isolates were re-analyzed, the scores from the repeats only were used in the analysis, not averaged. N=1 was used throughout this study (Line 164).

18. No isotopes or salts were used in the spectral analysis. 

19. The data transferred from the Vitek MS acquisition station was analyzed by Vitek MS RUO v4.12 (Line 167). 

20. In this study, each peak was treated as an independent observation (Line 183). It was not necessary to apply any corrections for the multiple student t-testing to adjust the individual p value for each peak to keep the overall error rate to less than or equal to the user-specified p cutoff value.

21. RAPD procedures were revised (Line 207-216).

22. Bacteria isolates included in this study were isolated from clinical specimens including 7 (7.4%) pus/wound, 5 (5.3%)sputum, 20 (21.1%) urine, 54 (56.8%) blood, and 5 (5.3%) body fluids like ascitic fluid, bile, pleural fluid and 4 (4.2%) other specimens like catheter tips.

23. This study was approved by the Institutional Review Board (VGHKS18-CT8-07) (Line 143-144).

24. Dimensionality is the number of peaks and is count data.

25. The conclusion that Random forest algorithm outperformed than the other algorithm was based on the ability to correct classification. Comments discussing the repeatedly misclassified isolate are written in discussion (Line 320-324). The L1O cross validation procedure was used to assess the predictive ability of different learning algorithms. The tested data is not part of the training set used in the building model and served as a blinded sample (Line 312-319).

26. These three differentially expressed peaks needs to be analyzed and processed using a large series of bacterial strains in the future to see if they are potential biomarkers (Line 296-298).

27. We used all the five machine learning algorithms that were offered in the Mathematica software v. 12 (Wolfram Research, Champaign, IL).

28. Figures and Tables were cited in the text. 

29. The training set and testing set was described in Materials and method (Line 178-179).

30. The X and Y axis in Figure 3 were specified.

31. Reference formatting was corrected.

Reviewer #2

1. The corresponding author is a statistics expert and performed analysis and interpretation. 

The revised manuscript is attached. We hope the revised manuscript is in satisfactory form and we look forward to hearing from you again soon.

Sincerely yours, 

Fu-Chuen Chang

Professor

Department of Applied Mathematics

National Sun Yat-sen University

70 Lien-hai Rd. 

Kaohsiung 804, Taiwan

---

## [Editor Report · Decision Letter 1]

16 Jan 2020

Detection of carbapenem-resistant Klebsiella pneumonia on the basis of matrix-assisted laser desorption ionization time-of-flight mass spectrometry by using supervised machine learning approach

PONE-D-19-26377R1

Dear Dr. Chang,

We are pleased to inform you that your manuscript has been judged scientifically suitable for publication and will be formally accepted for publication once it complies with all outstanding technical requirements.

With kind regards,

Joseph Banoub, Ph,D., D. Sc.

Academic Editor

PLOS ONE

Additional Editor Comments (optional):

This is to inform you that your manuscript has now been accepted for publication in PLOS one.
---

## [Editor Report · Acceptance letter]

24 Jan 2020

PONE-D-19-26377R1 

Detection of carbapenem-resistant *Klebsiella pneumonia* on the basis of matrix-assisted laser desorption ionization time-of-flight mass spectrometry by using supervised machine learning approach 

Dear Dr. Chang:

I am pleased to inform you that your manuscript has been deemed suitable for publication in PLOS ONE. Congratulations! Your manuscript is now with our production department. 

With kind regards,

on behalf of

Dr. Joseph Banoub 

Academic Editor

PLOS ONE